# Interindividual variability in the benefits of personal sound amplification products on speech perception in noise: A randomized cross-over clinical trial

**Maxime Perron**[1,2]*, **Brian Lau**[2], **Claude Alain**[1,2,3,4]

**1** Department of Psychology, University of Toronto, Toronto, Ontario, Canada, **2** Rotman Research Institute, Baycrest, Toronto, Ontario, Canada, **3** Institute of Medical Sciences, University of Toronto, Toronto, Ontario, Canada, **4** Music and Health Science Research Collaboratory, University of Toronto, Toronto, Ontario, Canada

* mperron@research.baycrest.org

**Data Availability Statement:** Individual data and the analysis scripts are available from the Borealis database (https://doi.org/10.5683/SP3/HTMDLI).

## Abstract

### Objective

The aging population is prone to hearing loss, which has several adverse effects on quality of life, including difficulty following conversations in noisy environments. Personal Sound Amplification Products (PSAPs) are a less expensive, over-the-counter alternative to traditional, more expensive hearing aids. Although some studies have shown that PSAPs can mitigate hearing loss, the literature generally only addresses group differences without considering interindividual variability. This study aimed to 1) determine how PSAPs affect listening effort and speech perception in noise and 2) measure interindividual variability and identify contributing demographic and health factors.

### Design

We used a cross-over design in which all participants were assigned to each condition.

### Participants

Twenty-eight adults aged 60 to 87 years with normal hearing and mild hearing loss fulfilled the study requirements.

### Intervention

In one session, speech-in-noise perception tasks were performed without PSAPs, and in the other, the tasks were performed with bilateral PSAPs. The two sessions were separated by one week, and the order of the sessions was balanced across participants.

### Main outcome measures

In both sessions, participants performed the Quick speech-in-noise test and a word discrimination task in noise, in which their self-reported listening effort was measured.

**Funding:** This work was supported by grants to CA from the Natural Sciences and Engineering Research Council of Canada (NSERC, https://www.nserc-crsng.gc.ca) [grant number RGPIN-2021-02721] and the William Demant Foundation (https://www.demant.com) [grant number 20-1260]. MP was funded by a Canadian Institutes of Health Research (CIHR) graduate scholarship. The funders had no role in study design, data collection and analysis, decision to publish, or preparation of the manuscript.

**Competing interests:** The authors have declared that no competing interests exist.

## Results

PSAPs use improved speech perception in noise in both tasks and reduced listening effort. There was considerable variability between individuals, with approximately 60–70% of participants showing benefit. Age, hearing and cognitive status were significant predictors of the benefits.

## Conclusion

Not all individuals may benefit from the effect of PSAPs to the same extent at their first use, and this depends on specific health and demographic factors, particularly age, hearing, and cognitive status. These results underscore the importance of demographic and health factors in assessing the benefits of hearing amplification in older adults.

## Trial registration

ClinicalTrials.gov, NCT05076045.

## Introduction

Hearing loss is highly prevalent among older adults. According to the World Health Organization, one-third of people over the age of 65 have debilitating hearing loss worldwide [1]. Hearing loss in older adults has been linked to social isolation and increased levels of depression and anxiety [2, 3], a higher risk of falls, hospitalization, mortality, and healthcare costs [4, 5]. In addition, older adults with hearing loss are more likely to experience cognitive decline and cognitive impairment [6, 7]. Finally, hearing loss is associated with communication difficulties, particularly understanding speech in a noisy environment, which is one of the characteristic symptoms of age-related hearing loss [8].

The traditional method of treating mild to moderate hearing loss is through hearing aids, which are sound amplification devices adjusted by audiologists to the level and specificity of the person's hearing loss. Chien and Lin [9] estimated that one in seven Americans over the age of 50 who have hearing loss use hearing aids. In a recent large cohort study of French adults, only 37% of people with hearing loss reported using hearing aids [10]. In Belgium, 38% of adults aged 80 and over with moderate hearing loss are underserved in the hearing care [11]. Considering that hearing loss and its adverse effects can potentially be mitigated by wearing hearing aids, these statistics highlight the importance of understanding why many people with hearing loss do not seek treatment.

Market research and surveys have listed the cost of hearing aids as the main reason older adults do not seek help for their hearing loss [12, 13]. According to a recent study on the affordability of hearing aids, an average cost of $2,500 (USD) for a hearing aid would be a catastrophic expense for 77% of Americans [14]. The authors suggested that reducing the cost of hearing aids to $500 or $1,000 would alleviate affordability issues for many people with hearing loss. However, it is important to acknowledge that hearing aid cost may not be the only issue [for a discussion, see 15]. Other factors, like attitudes towards hearing loss, may also play a role. Nonetheless, lowering the cost of hearing aids or finding affordable alternatives remains crucial to improving accessibility and reducing barriers to treatment.

In recent years, there has been an upsurge in the popularity of personal sound amplification products (PSAPs) and over-the-counter (OTC) hearing aids as affordable alternatives to

traditional hearing aids. Both types of devices are readily available and may be bought by customers without the help of a healthcare professional. There are, however, differences between PSAPs and OTC hearing aids. OTC hearing aids are medical devices that must comply with Food and Drug Administration (FDA) regulations and are used to treat mild to moderate hearing loss in people 18 years of age and older. The FDA established this new category of hearing aids to improve consumer access to hearing aids by enabling their purchase over-the-counter, without a prescription [16]. PSAPs are electronic devices not subject to FDA regulation and intended for people of all ages who want hearing amplification. Since PSAPs are categorized as consumer electronics rather than medical equipment, their product quality variability is likely higher than that of OTC hearing aids. Even though they are not designed to treat hearing loss, many PSAPs may adjust sound amplification to a person's hearing thresholds. Prices for PSAPs range from $20 to $500, making them readily available to individuals who may not have the means or insurance coverage to obtain hearing aids. PSAPs, therefore, offer an additional benefit in terms of accessibility and affordability. It is estimated that approximately 1.2 million people in the United States use PSAPs to compensate for their hearing loss [17], raising the importance of studying their effects.

Although PSAPs are gaining popularity, their appreciation and benefits on hearing ability are still equivocal. In a study of customer opinions, the authors identified many customers' negative impressions across all types of PSAPs studied, with more positive comments expressed about high and medium-cost products than about low-cost products [18]. This suggests that high and medium-cost products could be more effective than low-cost products. A recent meta-analysis of five studies comparing the use of PSAPs and conventional hearing aids on the hearing in noise test did not observe a difference between the PSAPs and hearing aids when pooling the groups. This suggests PSAPs might be as beneficial as conventional hearing aids in mitigating speech-in-noise difficulties [19]. However, some variability can be observed across studies, with some favouring PSAPs and others favouring conventional hearing aids. For instance, no significant improvement in speech-in-noise intelligibility has been reported when wearing PSAPs compared to hearing aids [20]. Moreover, the benefits of the PSAPs in the hearing in noise test are inconsistent across different types/brands of devices (i.e., hearing aid and PSAPs) and levels of hearing loss [21]. The benefits of PSAPs are limited to people with mild to moderate hearing loss, with little gain for more severe hearing loss [21, 22]. Even though PSAPs can mitigate hearing loss, the literature typically only discusses group differences without considering interindividual differences. It is unknown if the devices are helpful to everyone or if there are interindividual criteria that may predict who will benefit. This is important because not everyone uses and benefits from PSAPs similarly. Earlier PSAPs studies have only considered the severity of hearing loss as a predictor of the performance [21, 22]. However, several other variables, including sex, age, cognitive status, self-reported hearing, mental health, and physical health, have been associated with hearing loss [23] and may affect the effectiveness of PSAPs. This idea is also supported by research on adult hearing aid users, which revealed that the benefits of the devices for speech intelligibility were mainly related to younger age, better working memory, and milder hearing loss [24–26]. It is possible that these factors also predict who will benefit from PSAPs. Notably, interindividual variability may be higher with PSAPs because a person must fit them without assistance. As these devices gain popularity, it is essential to investigate their benefits and understand the interindividual variability in benefits.

This study aimed to recruit older adults with different hearing profiles to determine whether using PSAPs is associated with improved speech perception in noisy environments compared with an unaided condition. None of the participants had been diagnosed with hearing loss, but they were keen to explore the potential benefits of PSAPs for their hearing needs.

PSAPs are primarily designed for people who may have difficulty hearing in specific situations, even if their hearing is normal for their age. By including people with different hearing sensitivity as measured with pure tone audiometry, the study aimed to examine the effectiveness of PSAPs in improving the hearing experience of people who may have difficulties in certain listening situations, even if they have not been clinically diagnosed with hearing loss. We hypothesized that using PSAPs improves speech perception in noise, as measured by task performance and self-reported listening effort. This improvement would be predicted by hearing and cognitive status.

## Material and methods

### Participants

Recruitment took place from March 2022 to September 2022. Participants were recruited from Baycrest's participant database and through advertisements and word of mouth. The recruitment process of the participants is summarized in Fig 1. A total of 160 participants were contacted (no response = 49, not interested or Covid-19 concerns = 39), of whom 72 were assessed for eligibility. Of the 72, 22 did not meet inclusion criteria (tinnitus = 10, hearing aids = 12), 14 declined to participate, and four had other reasons. A total of 32 participants were recruited to participate in this study. Four participants were unable to finish the study for personal ($n = 2$) and medical ($n = 2$) reasons, resulting in a sample size of 28 participants (19 women, nine men). This sample size was determined based on previous research examining the effects of PSAPs. Previous studies did not report effect sizes or descriptive statistics between groups, making it difficult to conduct an accurate power analysis.

Participants ranged in age from 60 to 87 years (mean age 73.18, SD = 6.38 years). They all completed a phone interview to verify eligibility. The following criteria were required for inclusion: being right-handed, 60 years of age or older, and having English as a first language (or having learnt it before age five). The exclusion criteria included language impairment,

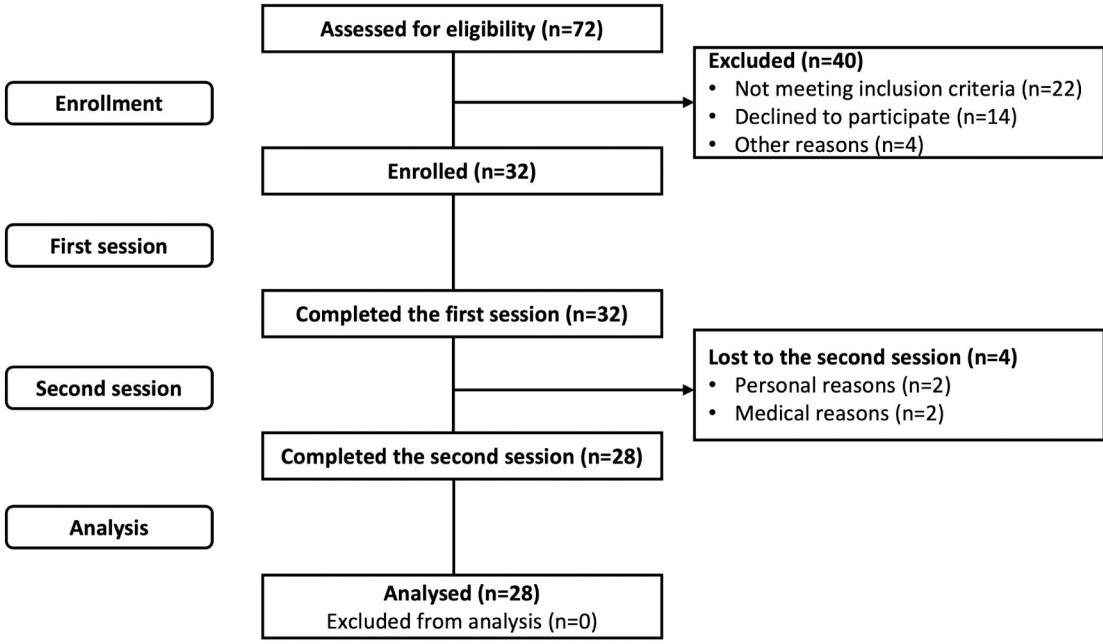

**Fig 1. CONSORT flow diagram.**

significant medical or neurocognitive conditions (such as dementia) or interventions that might affect cognitive function, untreated visual impairment, tinnitus or otologic disorders, known cerebrovascular disease, past or present experience with any hearing device, and contraindications to the PSAPs as described in the user manual.

This study was approved by the Baycrest Research Ethics Board (project #21–32) and is registered at ClinicalTrials.gov (NCT03008174). All participants gave written informed consent before the start of the experiment and received financial compensation.

## Procedure and design

The CONSORT checklist and the study protocol are provided in S1 and S2 Files, respectively.

All participants attended two sessions of approximately three hours each, with at least one week between them. At the start of the first session, participants completed several assessments, including the Montreal Cognitive Assessment (MoCA) [27], Speech, Spatial and Qualities of Hearing (SSQ) questionnaire [28], and a pure tone audiometry evaluation. In both sessions, they completed the 15-item version of the Geriatric Depression Scale (GDS) [29] and the 10-item version of the Geriatric Anxiety Inventory (GAI) [30]. Additionally, they performed two speech-in-noise perception tasks, the QuickSIN test, and a word discrimination task in three background noise intensities. Brain activity was recorded using electroencephalography during the word discrimination task (not reported here).

A randomized cross-over design was employed. The tasks were carried out in two sessions, one without PSAPs and the other with bilateral PSAPs. The order of the sessions was counterbalanced between participants using the block randomization method with blocks of three by the experimenters (MP, BL). The order of sessions was determined by the recruitment order (e.g., the first three participants performed the tasks without PSAPs first, then the next three with PSAPs, and so on). As a result, the order was made regardless of the participant's health. Thirteen participants performed the tasks without PSAPs first, while 15 participants completed the tasks with PSAPs first. All procedures took place at Baycrest's Rotman Research Institute in a double-walled soundproof room. Participants were given breaks during testing if necessary. The study design remained unchanged throughout the experiment.

## Hearing thresholds

Pure-tone thresholds in decibel (dB) hearing level (HL) were measured with a calibrated clinical audiometer (GSI 61, Grason-Stadler, USA). The following frequencies: 0.25, 0.5, 1, 2, 3, 4, 6, and 8 kHz were assessed in each ear separately. None of the participants had been diagnosed with hearing loss by an audiologist. Audiograms are provided in Fig 2. The peripheral hearing was operationalized as the pure tone average threshold (PTA) at 0.5, 1, 2 and 4 kHz of the better ear (better ear PTA4). According to this measure, 13 participants met the criterion for normal hearing (better ear PTA4 < 20 dB; mean = 13.46 ± 4.02, range: 5.00–18.75), and 15 participants met the criterion for mild hearing loss (better ear PTA4 between 20 and 49 dB; mean = 31.33 ± 6.11, range: 21.25–40.00) [31, 32]. Examining the PTA4 for each ear, 19 participants have hearing thresholds corresponding to the hearing loss criterion in at least one ear.

Line graphs present an overview of participants' hearing thresholds for the right ear (right panel) and the left ear (left panel). Each light gray line represents one participant. The black line represents the average thresholds of all participants.

## Amplification devices and fitting

Participants were fitted with bilateral PSAPs (CS50+, Sound World Solutions, Park Ridge, IL, USA). We used this device because it is one of the most studied PSAPs in the literature, making

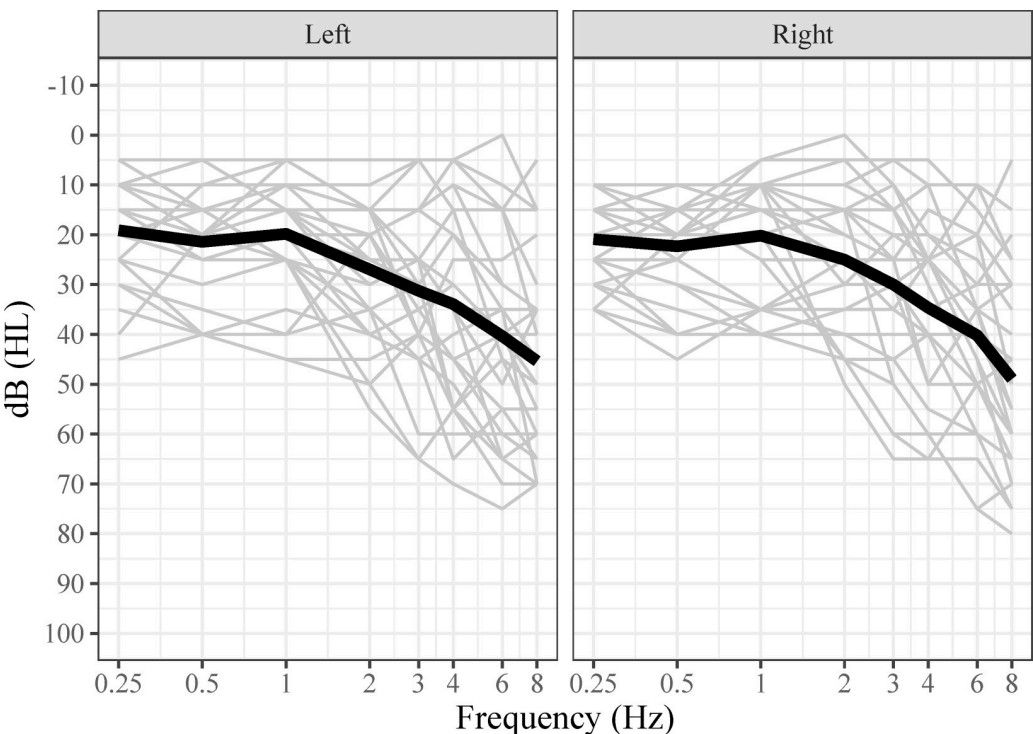

**Fig 2. Hearing thresholds.**

it easier to integrate our results with the results of previous studies. It is a high-end behind-the-ear sound amplifier with a 16-channel compressor, 16-channel noise reduction, feedback suppression, and a dual-directional microphone that is more sensitive to sounds from the front and less sensitive to sounds from the back. The price is approximately USD300 dollars per device, which is still significantly lower than traditional hearing aids.

The participants fitted the bilateral PSAPs themselves. They first selected the most appropriate ear tips from the three sizes (small, medium and large) according to their preferences and comfort level. They then took a hearing test in the PSAPs. The PSAPs were connected through Bluetooth to a mobile application on an iPhone (Apple, Cupertino, CA, USA) called Customizer by Sound World Solutions. Participants pressed a button when they heard a tone in one of the devices. This test is similar to a traditional pure-tone audiometry test in which pure tones of different frequencies are played in each ear separately. Instructions on how to conduct the test were provided in the iPhone app. No guidance was given to participants during the test, as these devices are intended to be fitted at home without the help of a healthcare professional. Although participants received no guidance, they were supervised during the fitting. The majority of participants were able to fit the devices themselves. However, on a few occasions, participants received assistance when encountering difficulties. Occasional guidance was provided to ensure that we were measuring the effectiveness of the devices rather than the participants' ability to fit them. After the hearing test, the iPhone app automatically sets the gain prescription (amplification) according to the in situ audiometric assessment results using the NAL-NL2 prescription [33]. The researchers could not access the test results because the app was a commercial product.

It should be noted that we did not use objective methods to test the PSAPs fitting process. In other words, we did not assess the consistency of audibility or the extent to which the

amplification matched the NAL-NL2 targets. Instead, the study relied on participants' subjective feedback to confirm the effectiveness of PSAPs. Indeed, after the fitting, participants were asked if they heard a change in amplification, and the experimenter changed the amplification level to confirm that the devices were working properly. Finally, the devices were put into "restaurant mode" to activate the dual directionality of the microphone and were set to the amplification level determined by the app. These options were retained for each task.

## Outcome measures

This study included three primary outcome measures and one secondary outcome measure. The three primary outcome measures included 1) accuracy on the Quick Speech-In-Noise (QuickSIN) [34], which measures the ability to perceive sentences in noise, and 2) accuracy and 3) reaction time on the word discrimination task, which measures the ability to discriminate speech sounds in noise. The secondary measure was a self-reported listening effort on the word discrimination task.

**QuickSIN.** Speech comprehension in noise was measured using the QuickSIN test [34]. The QuickSIN test contains several lists of six sentences with five keywords per sentence. The sentences are presented, and participants are asked to repeat them aloud. All sentences are presented in a four-talker babble noise. The sentences are presented at pre-recorded signal-to-noise ratios (SNRs) that decrease in 5 dB steps from 25 (very easy) to 0 (extremely difficult). The SNRs used are 25, 20, 15, 10, 5, and 0. For each correctly recalled keyword, participants receive one point out of a total of 5 per sentence. The total score of each list is calculated (for a total of 25), and an SNR loss score is calculated (25.5—score); then, the scores of all lists are averaged to give a final score, with lower scores indicating better speech-in-noise ability. In this study, four different lists were presented in the first and second sessions. During one session, lists 1, 2, 6, and 7 were presented, and during the other session, lists 3, 4, 8, and 9 were presented. The order of presentation of the lists for the two sessions was counterbalanced among the participants to avoid any differences in difficulty between the lists, if any. Also, the practice effect was controlled by counterbalancing the order of sessions between participants.

**Word discrimination task.** Phonological AX discrimination in noise was measured using a task similar to that developed by Tremblay and colleagues [35–37]. It involves discriminating Canadian-English pairs of monosyllabic Consonant-Vowel-Consonant (CVC) words. All words were selected from the Massive Auditory Lexical Decision (MALD) database [38], which contains stimulus records for 26,793 words and 9592 pseudowords recorded by a 28-year-old phonetics student from Western Canada. The pairs were either identical (e.g., /tap/-/tap/) or different (e.g., /bat/-/pat/). In 50% of the trials, the pairs were identical. When they were different, the difference could be located either on the onset (i.e., the first consonant) (33% of different trials), the nucleus (i.e., the vowel) (33% of different trials) or the coda position (i.e., the last consonant) (33% of different trials). On every trial, the words were presented simultaneously with a multi-talker's babble noise of a large group of people (~30) talking in a large, open room (https://freesound.org/people/mefrancis13/sounds/210611/). The audio file for the noise was segmented into 5-second segments that were played during the entire trial.

Three different signal-to-noise ratios (SNRs; +3 dB, 0 dB and -3 dB), calculated as the ratio of pressure signal to pressure noise, were utilized in the study. The audio files for the words were edited using PRAAT script to normalize root-mean-square (RMS) intensity at 70 dB sound pressure level (SPL). The audio files for noise were edited to normalize the RMS intensity to 67, 70, or 73 dB SPL. Thus, word intensity was constant, while noise intensity varied across trials to mimic a real-world situation. The sound level was adjusted at 70 dB SPL for all participants.

To utilize the dual-microphone directionality feature of the PSAPs, words were presented simultaneously through two speakers (Control 1 Pro, JBL, Los Angeles, CA, USA) placed in the front corners, while background noise was played through two speakers in the back corners of the soundproof room. This configuration was used to create a complex immersive auditory experience. All speakers were located approximately 1.5 meters from the participant's head, with a height of 1.2 meters and an angle of 45˚. To ensure that the participant remained seated and looked straight ahead without turning slightly toward the speakers, participants were instructed to focus on a fixation cross on a computer screen throughout the task. Video monitoring was used to ensure that participants complied with task instructions.

Two versions of the same task were created, one for each testing session. In both versions, the material was the same, except that the order of the words in each pair was reversed, and the pairs were presented in different SNRs. In both sessions, the experiment began with a practice block of 16 trials. The main task was divided into three blocks of 10 minutes each, with a short break between blocks. A total of 306 pairs were presented in both sessions, including 102 pairs in each SNR condition. In each trial, the babble noise began to play for 1000 milliseconds (ms), and then the two words were presented with an inter-stimulus interval of 300 ms. After the words were presented, participants were asked to determine whether the words were the same or different using a response box (Cedrus, model RB-530). A white fixation cross centred on a dark gray background was presented during the words, followed by a green question mark (?) to instruct participants to respond. The inter-trial interval was 1000 ms. Participants had a maximum of 3 s to respond. All stimuli and experiment files are available at https://doi.org/10.5683/SP3/HTMDLI/.

**Self-reported listening effort.** Participants used a seven-point Likert scale to rate the listening effort required to complete each of the three blocks of the word discrimination task. The scale used was the one adopted by Johnson et al. [39]. The specific question was: "Using the scale in front of you, can you estimate how much effort it took you to understand the words in the presence of background noise? If you think that the amount of effort was between two numbers on the scale, it is fine for you to pick a fraction," with number 1 corresponding to "No effort" and number 7 corresponding to "Extreme effort."

## Demographic and health factors

Demographic factors included age and biological sex. Health factors included cognitive functioning, depression, anxiety, self-rated health, and subjective measure of hearing. Cognitive functioning was measured using the Montreal Cognitive Assessment Hearing Impairment (MoCA-HI) [27]. Depression and anxiety were measured in both sessions using the short versions of the GDS [29] and GAI [30], respectively. Self-rated health was assessed using the following question: 'In general, how would you rate your health today?' The response options included 'very good', 'good', 'moderate', 'bad', or 'very bad' [40]. Finally, the self-reported ability to listen to speech in quiet and in noise was documented using the Speech scale of the Speech, Spatial and Qualities of Hearing Scale [28]. A summary of the participants' information is provided in Table 1.

## Analyses

Individual data and the analysis scripts are publicly available at https://doi.org/10.5683/SP3/HTMDLI. Data were analyzed using R version 4.1.1 [41] in R Studio [42]. Outliers (values more than 3 interquartile range) were removed for each dependent variable. At most, one outlier was removed per dependent variable. Next, the variable distributions were visually inspected using histograms. For each task, two sets of analyses were conducted. The first was

**Table 1. Demographic and health characteristics of study participants (n = 28).**

| Variable | Mean | SD | Range |
|---|---|---|---|
| Age | 73.18 | 6.38 | 60–87 |
| Education (years) | 16.5 | 2.52 | 12–22 |
| MoCA-HI[a] (/30) | 27.57 | 2.13 | 22–30 |
| GDS[b] (/15) [without] | 1.46 | 1.86 | 0–8 |
| GDS (/15) [with] | 1.14 | 1.6 | 0–6 |
| GAI[c] (/30) [without] | 2.46 | 2.4 | 0–10 |
| GAI (/30) [with] | 2.89 | 2.27 | 0–11 |
| Self-rated health[d] (/5) | 1.61 | 0.69 | 1–3 |
| Speech SSQ score[e] (/10) | 7.97 | 1.36 | 4.71–10 |
| Left ear PTA4[f] | 25.54 | 11.3 | 5–47.5 |
| Right ear PTA4 | 25.54 | 11.04 | 7.5–43.75 |
| Better ear PTA4 | 23.04 | 10.44 | 5–40 |

Notes.

[a.] Montreal Cognitive Assessment Hearing Impaired. Higher scores indicate better general cognitive functions.

[b.] 15-item version of the Geriatric Depression Scale. Higher scores indicate a more depressed state. Scores labelled "without" were collected in the session when participants did not utilise PSAPs, whereas scores labelled "with" were collected in the session where participants did.

[c.] 10-item version of the Geriatric Anxiety Inventory. Higher scores indicate a more anxious state. Scores labelled "without" were collected in the session when participants did not utilise PSAPs, whereas scores labelled "with" were collected in the session where participants did.

[d.] Self-rated health was documented using this question: "In general, how would you rate your health today" with the possible choices being "very good," "good," "moderate," "bad," or "very bad."

[e.] Speech scale of the Speech, Spatial and Qualities of Hearing Scale composed of 14 items documenting, among other things, the self-reported ability to listen to speech in quiet and in noise.

[f.] Pure tone average threshold measured in decibels at 0.5, 1, 2, 4 kHz.

to investigate the difference between sessions (i.e., without, with PSAPs) (Objective 1). The second was to determine if demographic and health factors could explain interindividual variability in using PSAPs (Objective 2). For all analyses, an alpha level of 0.05 was used to determine the statistical significance

**Difference between session.** For the **QuickSIN task**, the SNR loss score for both sessions was not normally distributed. Therefore, the performance of the QuickSIN for the two sessions was compared using the Wilcoxon signed rank test on paired samples using the wilcox_test function in the rstatix package version 0.7. The Wilcoxon effect size (r) is reported.

For the **word discrimination task**, the two main variables of interest were accuracy (in percent) and reaction time (RT) (in ms). RTs were log-transformed. Data were analyzed using linear mixed-effects (LME) models separately for each dependent variable. Each LME model was fitted using the lmer function in the lme4 package version 1.1.27. Models included Session (without, with) and SNR as within-subject factors. Participants were included as a random factor.

For **self-reported listening effort**, because the Likert scale is ordinal in nature, the listening effort score was analyzed using a cumulative link mixed model. The model was fitted using the clmm function in the ordinal package version 2019.12.10. A logit link function with flexible thresholds was used. The model included Session (without, with) and Block (1, 2, 3) as within-subject factors. Participants were included as a random factor.

**Factors contributing to interindividual variability.** For each behavioural measure that showed differences between sessions in the previous set of analyses described, we performed

an additional LME model to identify predictors of the difference. Given the limited sample size, we focused on four predictors: age, MoCA-HI score, best ear PTA4, and the SSQ speech scale. The choice of predictors was based on the hearing aid literature [24–26], which shows that these variables are associated with hearing amplification benefits. We additionally included the self-reported measure of hearing because it has been shown that objective and subjective measures of hearing do not correlate strongly and may reveal different aspects of hearing ability [e.g., 23].

Prior to the analyses, statistical assumptions were tested, including normality, residuals' homoscedasticity, and influential outliers. If the distribution was considered normal, the LME method was chosen. If not, a generalized LME approach was used. Each model had the same structure. The models included the behavioural measure as the independent variable and the interactions between Session and each predictor as fixed effects (Performance ~ Session * Age + Session * MoCA-HI + Session * Better Ear PTA, Session * Speech SSQ). Participants were included as a random factor. All predictors were mean-centered. At most, two outliers were removed per model. Unstandardized ($B$) and standardized ($\beta$) coefficients are reported.

## Results

### Difference between session

**QuickSIN.**   Descriptive statistics from the QuickSIN revealed that 18 of 28 (64.3%) participants had better performance (i.e., lower SNR loss score) with the PSAPs than without (for more details, see Table 2). A Wilcoxon signed-rank test showed that PSAPs significantly improved speech-in-noise perception with a large effect size, $V = 71$, $p = 0.005$, $r = 0.53$. The PSAPs are associated with lower median SNR loss score (Fig 3), which suggests better speech-in-noise ability. An exploratory analysis of only those who met the criterion of hearing loss in one ear (N = 19) showed consistent results, $V = 34.5$, $p = 0.03$, $r = 0.50$.

**Word discrimination task.**   For accuracy, descriptive statistics from the word discrimination task revealed that 17 of 28 participants (60.7%) had higher accuracy (all three SNR combined) with the PSAPs (for more details, see Table 2). The LME model for accuracy revealed significant main effects of SNR and Session with no significant interaction (Table 3A). Pairwise comparisons on estimated marginal means were conducted to decompose the main effects. The main effect of SNR revealed a decrease in accuracy with a decrease in SNR (Fig 4A) (SNR +3 dB: $M = 87.8\%$, SNR 0 dB: $M = 85.6\%$, SNR -3 dB: $M = 80.5\%$), with a significant difference between all three conditions (SNR +3 dB—SNR 0 dB: $t(135) = 3.13$, $p = 0.007$; SNR +3 dB— SNR -3 dB: $t(135) = 10.46$, $p < 0.001$; SNR 0 dB—SNR -3 dB: $t(135) = 7.33$, $p < 0.001$). The main effect of Session revealed a slight but significantly higher accuracy in the session with the PSAPs ($M = 85.2\%$, 95% CI = [83.2, 87.4]) compared to the session without the PSAPs ($M = 84.0\%$, 95% CI = [81.9, 86.0]) (Fig 4B), $t(135) = -2.18$, $p = 0.03$. An exploratory analysis of

**Table 2. Number of participants showing improved, decreased, or no change in task performance with and without PSAPs.**

| Task | Outcome | Change in performance | | |
|---|---|---|---|---|
| | | Better with PSAPs | Better without PSAPs | No Change |
| QuickSIN | SNR loss (/28) | 18 (64.3) | 9 (32.1) | 1 (3.6) |
| Word discrimination task | Accuracy (/28) | 17 (60.7) | 10 (35.7) | 1 (3.6) |
| | Reaction time (/27) | 12 (44.4) | 15 (55.6) | 0 (0) |
| | Listening effort (/28) | 19 (67.9) | 5 (17.9) | 4 (14.3) |

**Notes.** Values in parentheses indicate the percentage of participants demonstrating each type of performance change.

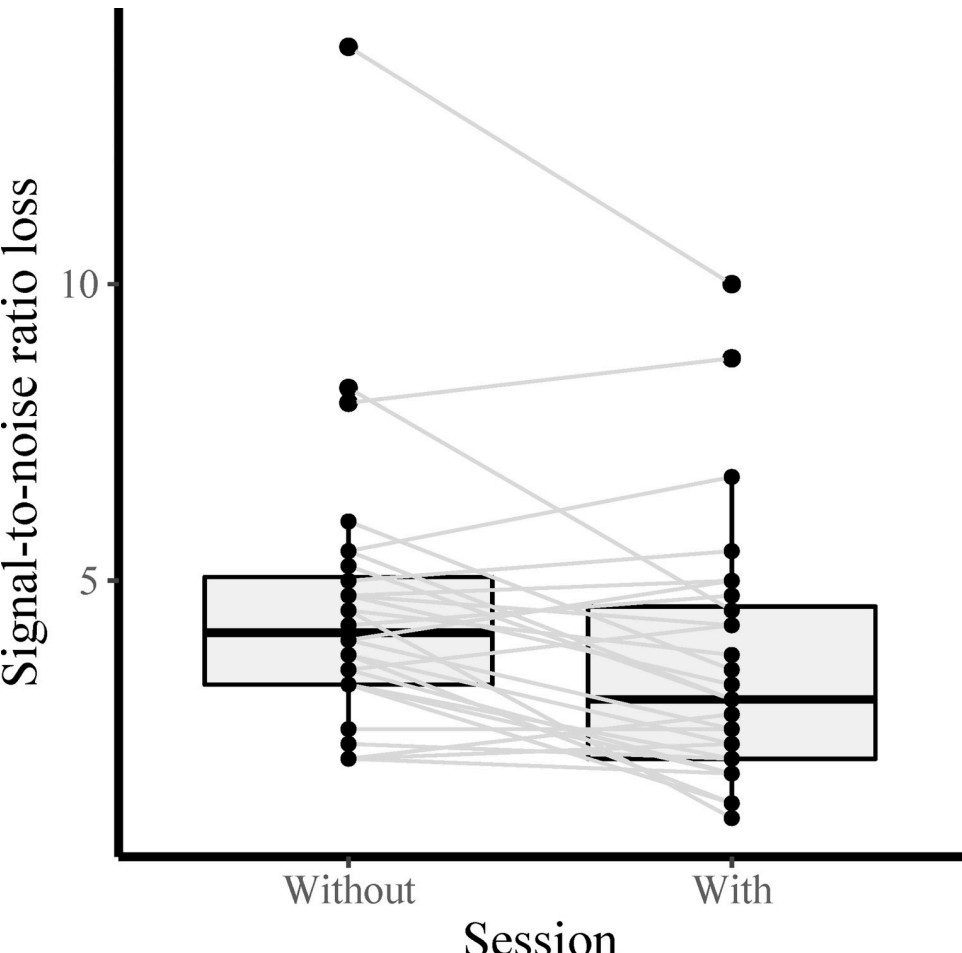

**Fig 3. Differences in signal-to-noise ratio (SNR) loss on the QuickSIN test between sessions with and without personal sound amplification products.** Each line represents an individual's data.

only those who met the criterion of hearing loss in one ear (N = 19) revealed no significant main effect of Session.

For RTs, descriptive statistics revealed that 12 of 27 participants (44.4%) (1 missing data) had faster RTs with PSAPs than without PSAPs (for more details, see Table 2). One outlier was removed (mean RTs > 2.5 s). The LME model for log-transformed RTs revealed only a moderate main effect of SNR (Table 3B). Bonferroni-corrected pairwise comparisons revealed an increase in RTs with a decrease in SNR (SNR +3 dB: $M$ = 3.68, SNR 0 dB: $M$ = 3.69, SNR -3 dB: $M$ = 3.72) (Fig 4C), with a significant difference only between SNR +3 dB and SNR -3 dB, $t$ (127) = -2.80, $p$ = 0.02.

**Self-reported listening effort.** Descriptive statistics of the reported listening effort revealed that 19 of 28 participants (67.9%) reported less effort (in all blocks) when using PSAPs (for more details, see Table 2). The cumulative link mixed model revealed main effects of Block ($\chi^2(2)$ = 7.70, $p$ = 0.02) and Session ($\chi^2(1)$ = 16.43, $p < 0.001$), with no significant interaction between factors. An exploratory analysis of only those who met the criterion of hearing loss in one ear (N = 19) showed a consistent main effect of Session $\chi^2(1)$ = 4.73, $p < 0.03$.

The probability of reporting each score for each block is shown in Fig 5A. The most likely score in all blocks was 4 out of 7 (i.e., moderate effort). Notably, the probability of reporting a

**Table 3. Results of linear mixed-effects models on word discrimination task performance.**

| Effect/Interaction | F | NumDF | DenDF | p | $\eta_P^2$ |
|---|---|---|---|---|---|
| **A. Accurary** | | | | | |
| Session | 4.76 | 1 | 135.0 | 0.03 | 0.03 |
| SNR | 57.63 | 2 | 135.0 | < 0.001 | 0.46 |
| Session:SNR | 0.02 | 2 | 135.0 | 0.98 | < 0.001 |
| **B. Reaction Time** | | | | | |
| Session | 1.16 | 1 | 126.9 | 0.28 | 0.01 |
| SNR | 4.46 | 2 | 126.4 | 0.01 | 0.07 |
| Session:SNR | 0.18 | 2 | 126.4 | 0.83 | 0.003 |

**Notes.** NumDF: degrees of freedom in the numerator, DenDF: degrees of freedom in the denominator, $\eta_P^2$ = partial eta squared.

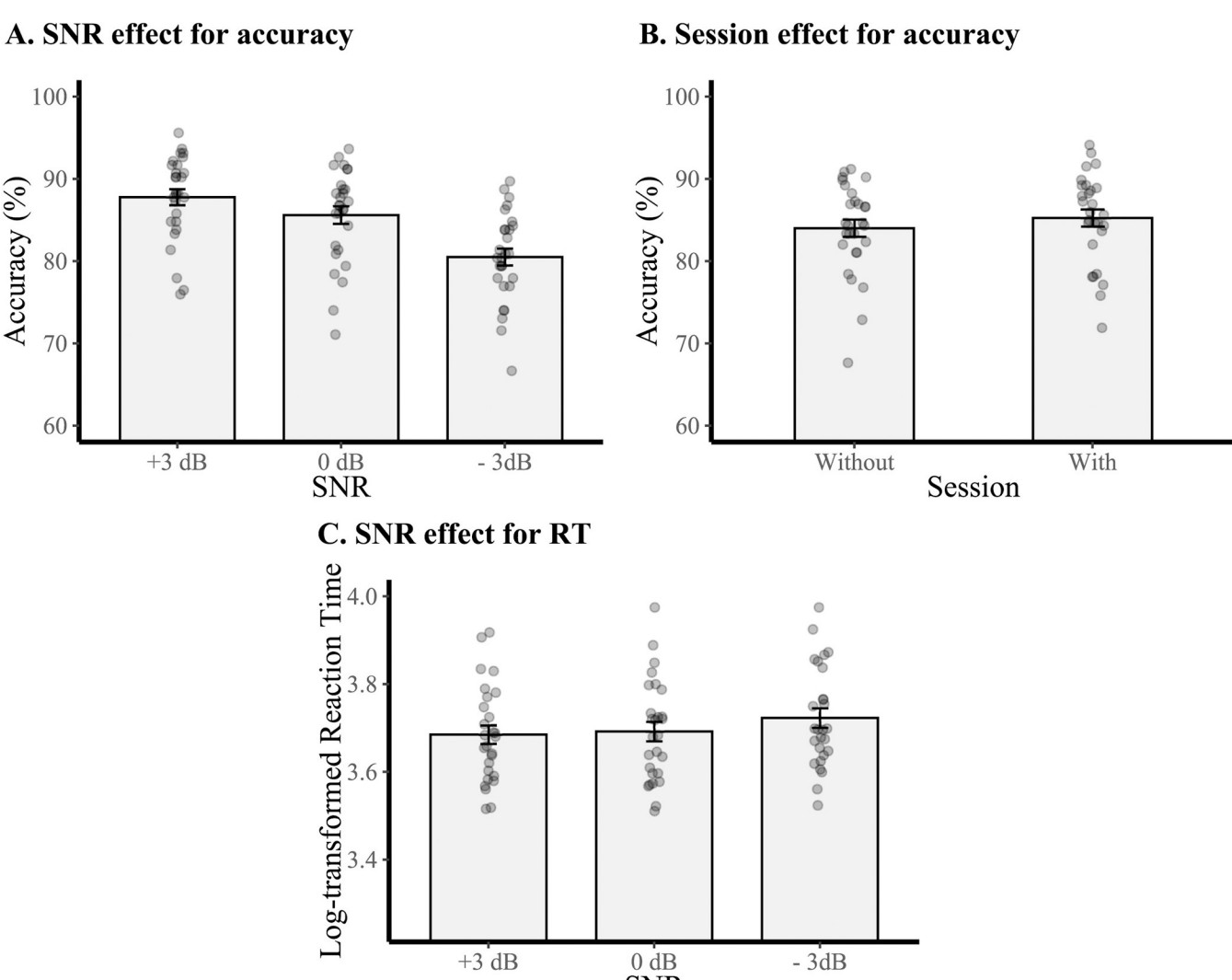

**Fig 4. Results for the word discrimination task (predicted values).** (A) The bar graph displays the main effect of signal-to-noise ratio (SNR) for accuracy. (B) The bar graph displays the main effect of session for accuracy. (C) The bar graph displays the main effect of SNR on reaction time. Each dot represents one participant. Error bars represent 95% confidence intervals of the mean.

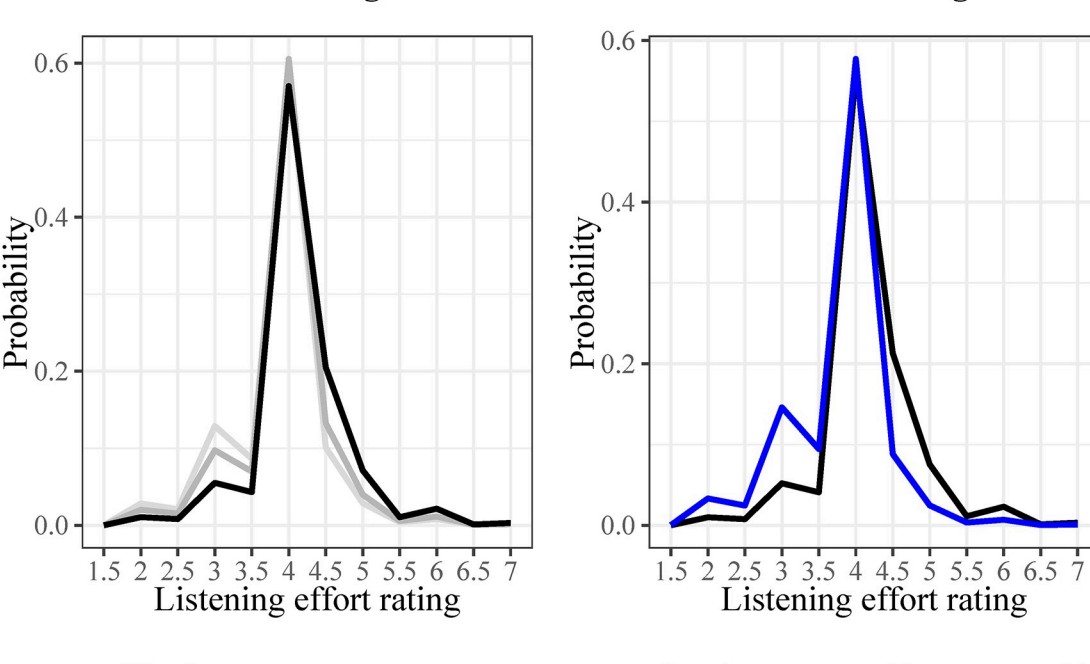

**Fig 5.** The graphs show the main effect of A) blocks and B) session on the probability of reporting each listening effort score during the word discrimination task.

small listening effort score (i.e., below 4) decreases, and the likelihood of reporting a large score (i.e., above 4) increases with the number of blocks. Pairwise comparisons on the average score class revealed an increase in listening effort with blocks (Block 1: $M$ = 5.67 [score between 3.5 and 4], Block 2: $M$ = 5.89 [score close to 4], Block 3: $M$ = 6.28 [score between 4 and 4.5]), with a significant difference only between Blocks 1 and 3 ($z$ = -2.66, $p$ = 0.02), suggesting greater effort at the end than at the beginning of the task.

The probability of reporting each score for each session (without, with PSAPs) is shown in Fig 5B. The most likely score in both sessions was 4 out of 7 (i.e., moderate effort). Participants were likelier to report a low listening effort score (i.e., below 4) and less likely to report a high score (i.e., above 4) when using the PSAPs. Pairwise comparisons on the average score class revealed that participants reported on average higher scores without PSAPs ($M$ = 6.32 [score between 4 and 4.5], 95% CI = [5.71, 6.92]) than with ($M$ = 5.59 [score between 3.5 and 4], 95% CI = [4.94, 6.22]), $z$ = 3.98, $p$ < 0.001.

### Factors contributing to interindividual variability

For each behavioural measure that showed differences between sessions (SNR loss score, accuracy on the word discrimination task and listening effort score), we performed an additional LME model to identify predictors of the difference.

For QuickSIN, a generalized LME model was used. Because the scores were positively skewed with a continuous positive range, the model was fitted using a gamma distribution and a log link function. The model revealed significant interactions between Session and Age, Session and Better ear PTA4, and Session and Speech SSQ (Table 4A). Age was a stronger positive predictor of QuickSIN performance when participants wore PSAPs than when they did not.

**Table 4. Results of linear mixed-effects models to identify predictors of session difference.**

|  | B | β | Std. Error | t | p | 95% CI |
|---|---|---|---|---|---|---|
| **A. QuickSIN** | | | | | | |
| Session:Age[Without] | -0.03 | -0.2 | 0.01 | -2.99 | < 0.01 | [-0.34, -0.07] |
| Session:MoCA[Without] | 0.03 | 0.06 | 0.03 | 1.05 | 0.29 | [-0.06, 0.19] |
| Session:PTA4[Without] | 0.01 | 0.15 | 0.01 | 2.17 | 0.03 | [0.01, 0.28] |
| Session:Speech SSQ[Without] | 0.10 | 0.14 | 0.05 | 2.28 | 0.02 | [0.02, 0.26] |
| **B. Accuracy (word discrimination task)** | | | | | | |
| Session:Age[Without] | 0.06 | 0.07 | 0.12 | 0.50 | 0.62 | [-0.22, 0.37] |
| Session:MoCA[Without] | -1.03 | -0.36 | 0.40 | -2.61 | 0.02 | [-0.63, -0.08] |
| Session:PTA4[Without] | 0.06 | 0.12 | 0.07 | 0.80 | 0.43 | [-0.19, 0.44] |
| Session*Speech SSQ[Without] | -0.31 | -0.08 | 0.51 | -0.61 | 0.55 | [-0.36, 0.19] |
| **C. Listening effort (word discrimination task)** | | | | | | |
| Session:Age[Without] | 0.00 | -0.01 | 0.03 | -0.04 | 0.97 | [-0.45, 0.43] |
| Session:MoCA[Without] | -0.02 | -0.05 | 0.09 | -0.27 | 0.79 | [-0.44, 0.34] |
| Session:PTA4[Without] | -0.01 | -0.11 | 0.02 | -0.53 | 0.60 | [-0.55, 0.32] |
| Session:Speech SSQ[Without] | -0.03 | -0.04 | 0.14 | -0.22 | 0.83 | [-0.43, 0.34] |

**Notes.** *B*: unstandardized coefficients, *β*: standardized coefficients, 95% CI: 95% confidence level.

Age therefore reduces the advantages of PSAPs (Fig 6A). Best ear PTA4 was a stronger positive predictor of QuickSIN performance when participants did not wear PSAPs than when they did. Participants with higher hearing thresholds benefited more from PSAPs (Fig 6B). Finally, the score at the Speech SSQ was a stronger negative predictor of QuickSIN performance when

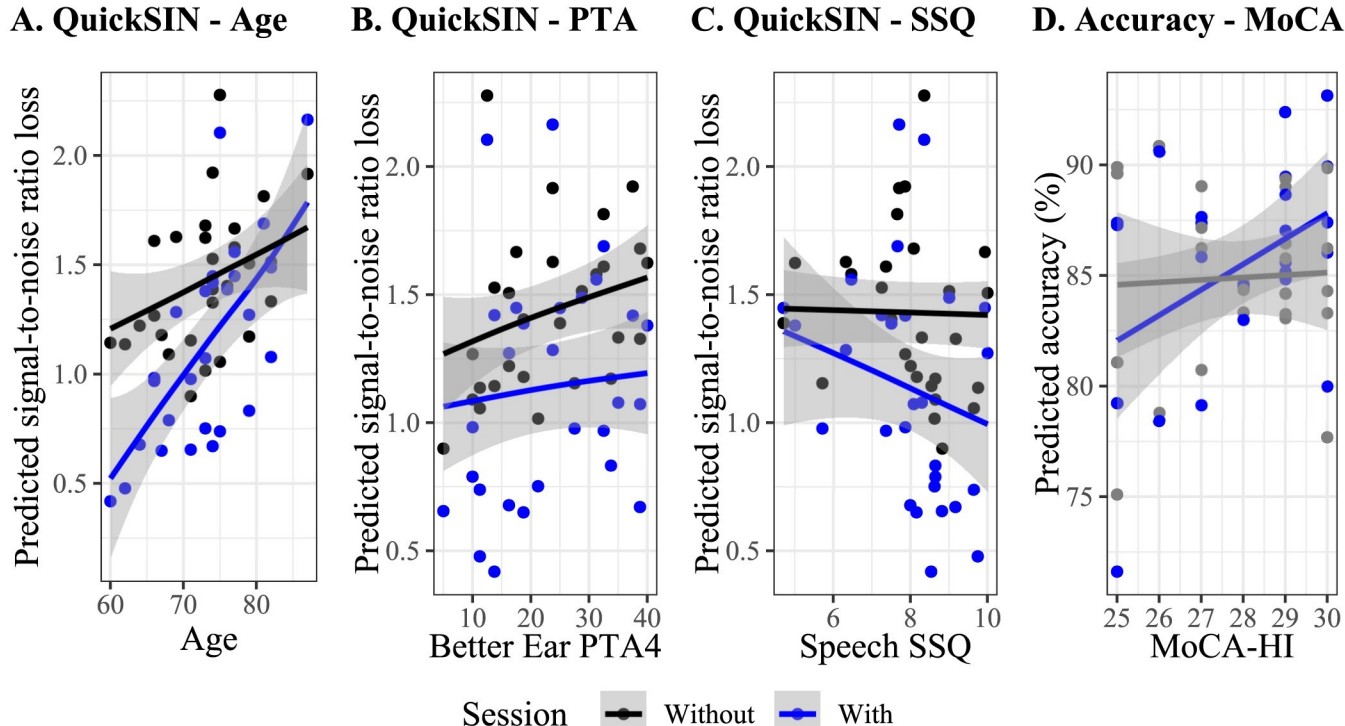

**Fig 6. Scatterplots display the interaction between session (without, with PSAPs) and demographic and health factors.** (A-C). Results for the QuickSIN. (D). Result for accuracy on the word discrimination task.

participants wore PSAPs than when they did not. Participants who reported superior listening skills benefited more from PSAPs (Fig 6C).

For the word discrimination task, an LME model revealed a significant interaction between Session and the MoCA-HI, with no other significant interactions (Table 4B). Cognitive scores were a stronger positive predictor of task performance when participants wore PSAPs than when they did not, with benefits increasing with higher cognitive scores (Fig 6D).

For the listening effort score, an LME model revealed no significant interaction between Session and the four factors (Table 4C).

## Discussion

Personal sound amplification products (PSAPs) and over-the-counter (OTC) hearing aids are becoming a new alternative for individuals with hearing loss. Although a growing body o advantages of using these devices at the group level, there has been limited focus on individual differences. Therefore, the full extent of their benefits remains unknown. To clarify the interindividual variability in benefits, the present study used a cross-over design to assess the beneficial effects of PSAPs on speech perception in noise and self-reported listening effort in two counterbalanced sessions. Consistent with previous research, this study supports that PSAPs use is associated with speech-in-noise benefits in terms of performance and listening effort. However, the benefits are small and only observed for some individuals.

### Effects of the PSAPs on speech perception in noise

In this study, we focused on speech perception in noise because difficulty following a conversation in a noisy environment is one of the most common complaints of older adults with and without hearing loss. Two different tasks were used to measure speech-in-noise perception. The first was the QuickSIN [34], a clinical hearing test that quickly and easily measures the ability to hear in noise. The second was a word discrimination task in noise that allowed us to measure the effect of PSAPs on phonological discrimination in noise and listening effort. In both tasks, the aided condition (i.e., with PSAPs) was associated with better performance than the unaided condition. Specifically, participants were more accurate in discriminating words in noise and repeating sentences under high noise levels in the QuickSIN test with the PSAPs. This is consistent with previous studies showing that PSAPs are associated with better listening ability than an unaided condition [21, 22, 43].

In the present study, PSAPs use was associated with reduced self-reported listening effort. According to the information degradation hypothesis [44], listening difficulties compromise the higher level of cognitive processing as resources are used for better auditory perception. Thus, more effortful listening situations place greater demands on executive functions and working memory, adversely affecting cognition. Our results suggest that hearing amplification may decrease listening effort, thereby potentially mitigating the negative impact of hearing loss on cognitive performance in the long term. However, more empirical data are needed to support this claim. Although we observed significant differences in performance between the aided and unaided conditions, the benefits identified were small, particularly for the word discrimination task, for which only a benefit of approximately 1% was identified, which may not be clinically meaningful. Across all dependent measures, we found that only 60–70% of individuals benefited from the hearing amplification, revealing substantial interindividual variability in using PSAPs. Nonetheless, these results are important because they show that PSAPs have a positive effect at the group level, as shown in other previous studies, but that there appears to be significant variability at the individual level. It is also worth mentioning that we found a significant benefit despite little exposure or practice with the devices and little

assistance with fitting them. This result suggests that older adults can adapt quickly to wearing PSAPs and benefit from the increased signal at the ear, despite only a three-hour exposure. More prolonged use would likely be more effective as individuals become more accustomed to the devices and the amplification. The amplification can be overwhelming. For this reason, conventional hearing aids are gradually adapted to the participants' hearing to allow them to get used to them. According to a recent study, the brain would quickly adapt to the amplification within two weeks. Still, it would take up to six weeks to fully interpret the amplified signals meaningfully [45]. Here, the PSAPs were not gradually adapted to hearing and were only worn for one session of three hours, which could explain the small yet significant benefits and inter-individual variability. A fitting procedure similar to conventional hearing aids could be used for these devices in future longitudinal studies.

Another factor that could explain the gain in speech-in-noise perception is that the PSAPs currently on the market might not be effective enough to provide clinically relevant benefits, as they do not all have the same features as conventional hearing aids. This would be consistent with a previous study that examined the impact of five different PSAPs and hearing aids on listening ability [21]. Their results showed that all PSAPs improved auditory processing, but the gains were significantly less than conventional hearing aids. At the opposite, a recent meta-analysis found no performance difference between PSAPs and conventional hearing aids by pooling results from five different studies [19]. Nevertheless, the studies included in the meta-analysis showed considerable variability, with some favoring PSAPs and others favoring hearing aids, which could be due to differences in the brands of PSAPs and hearing aids used, but also to the conceptual differences between PSAPs and hearing aids.

There are important conceptual differences to consider when comparing traditional hearing aids, OTC hearing aids and PSAPs. These differences relate to personalization, effectiveness, safety and affordability. Traditional hearing aids are medical devices that must be prescribed, are tailored to the specific needs of the user, and are regulated by the FDA to ensure their effectiveness and safety. Similar features are found in OTC hearing aids, but they do not require a prescription and can be adjusted by the users themselves, making them more widely available and less expensive. On the other hand, PSAPs are not subject to the same regulations as medical devices and may have fewer and less advanced features, which could reduce their effectiveness. While conventional hearing aids should remain the preferred choice for hearing care, the results of this study suggest that PSAPs may be a viable alternative for those who cannot afford both conventional or OTC hearing aids, as they appear to offer significant benefits compared to an unaided condition. The results of this study come in the context that conventional hearing aids are rarely reimbursed in full or even in part in most countries. If conventional devices were reimbursed, they would be a safe and effective option for treating hearing loss. In this case, PSAPs may still be an attractive option in some cases, such as for very mild hearing loss or when a person needs amplification under certain conditions. In line with this idea, the current study shows that individuals with diverse hearing profiles, even without any clinical diagnosis of hearing loss, can experience benefits from PSAPs, providing tangible evidence that these devices can help people who want to improve their auditory communication skills.

Future studies are, however, needed to compare different PSAP brands on speech perception and listening ability. However, this may be challenging as the market will likely grow in the coming years, making it challenging to compare the different types of PSAPs. Future studies should examine which parameters and settings are most beneficial on PSAPs, thus providing guidelines for those who opt for this option instead of conventional or OTC hearing aids. Here, we used high-end PSAPs with noise cancellation features, making our results only generalizable for this device and other devices with the same characteristics.

Notwithstanding, it is encouraging that most participants did better with PSAPs than without, even for a short period of use. There are no longitudinal studies to date exploring the effects of these devices on measures of speech perception and auditory processing. The next step would be to conduct longitudinal studies to explore the benefits of these devices on speech perception and compare these benefits to those of hearing aids.

## Interindividual variability

Finally, another factor that may contribute to the interindividual variability we examined in this study is the impact of demographic and health factors. Age-related hearing loss is a complex condition that does not affect all individuals equally. Some people with specific characteristics are more likely to have hearing loss, or report hearing loss than others. For example, in a recent study conducted by our group, we found significant differences in speech-in-noise ability between males and females, with males showing a higher loss than females [23]. We also observed that the likelihood of reporting hearing loss is affected by age, depression, anxiety, age, and self-reported health status. Other studies have also shown that the likelihood of having objective hearing loss measured by audiogram depends on demographic and health factors [e.g., 46]. Since demographic and health factors are associated with hearing loss, it is likely that these factors also affect how a person responds to hearing amplification. Research has shown that age, hearing thresholds, and executive function predict the benefits of conventional hearing aids on speech intelligibility. These factors are likely to predict interindividual differences in response to sound amplification with PSAPs, but this remains unknown.

Stepwise regression analyses were used to determine whether interindividual differences in performance with and without PSAP could be explained by age, hearing thresholds (Better ear PTA4), self-reported listening ability (Speech SSQ scale) and cognitive functioning (MoCA-HI). We found that the benefits of PSAPs can be explained by some demographic and health factors, at least for the PSAPs we employed in the current study and others with comparable features. Although these factors do not necessarily explain who will benefit most from these devices in the long term, they highlight important factors that future studies should consider when evaluating the impact of these devices on hearing and listening.

First, we observed that people who met the criterion for hearing loss, who were younger older adults and reported having good speech-listening skills, had higher benefits on the QuickSIN with the devices. The effect of hearing thresholds is not surprising because few studies show that hearing thresholds are a significant determinant of the impact of PSAPs on listening ability. The devices seem beneficial for people with mild-to-moderate hearing loss but not for severe hearing loss [21, 22]. The impact of age and self-reported listening effort is more surprising. This suggests that the benefits of PSAPs would decrease with age. Several scenarios can explain this effect and are worthy of future research. One possibility is that the oldest-old adults may have more difficulties with the technology than the youngest-old adults and may need additional help fitting the devices to their hearing thresholds. Another possibility is that the oldest-old adults may be more upset by amplification than the youngest-old adults, as they may be less resistant to change in their hearing. Oldest-old adults may have lived with their hearing loss for longer and, therefore, may have difficulties quickly adjusting to hearing amplification. This would be consistent with findings suggesting that older adults report less hearing loss because they consider it a normal part of aging. Thus, hearing amplification may be a big shock to them. Future studies should measure participants' discomfort or impression of the sound quality and effects of the devices. The impact of self-reported hearing loss is also interesting, as one would expect the opposite; people who report more hearing difficulty would benefit more from the devices than those with no hearing difficulty. However, it is well known

that self-reported and objective measures of hearing are not strongly correlated [e.g., 20, 23], meaning that individuals may not be aware of their hearing status. One hypothesis that might explain this outcome is that those who report more difficulty may be more negative about their condition and, thus, less optimistic about the usefulness of the devices. However, the effect of self-reported listening ability does not significantly explain the benefits but does help explain the interindividual variability in the regression model.

Another interesting result is that individuals with higher scores on the MoCA-HI benefitted more from the devices than those with lower scores. Here, all participants' MoCA-HI scores were within a range considered cognitively healthy, showing that even normal cognitive score variability can lead to changes in PSAPs effects. The difference in PSAPs uptake as a function of cognitive status suggests that individuals with lower cognitive scores may be more overwhelmed by amplification than those with higher cognitive functioning. For instance, individuals with lower cognitive functioning may have fewer cognitive resources to adjust for the effect of amplification. This would be consistent with the literature on conventional hearing aids showing that speech intelligibility in noise appears worse for listeners with low working memory capacity than for those with high capacity [24]. It has also been shown that older listeners with hearing loss and poor working memory are more sensitive to distortion caused by some hearing aid signal processing algorithms and noise [47]. It has been proposed that listeners with low working memory capacity can adapt less to rapid signal changes. However, since we observed the contribution of cognitive status only for the word discrimination task and not for QuickSIN, this suggests that it is true but only when the task is cognitively demanding.

Finally, for listening effort, no single factor significantly explains the variability in the benefits. Looking at the individual data, we observe that almost all participants reported lower listening effort with the devices than without the devices. This suggests that one mechanism by which devices can help improve listening ability is by reducing listening effort. However, participants may also report less listening with the device because they believe it should help them hear better. Future studies should consider supplementing subjective rating of listening effort with more objective metrics such as pupil dilatation [e.g., 48, 49].

## Limitations

In the present study, we focused on a single brand of high-end PSAPs. This choice limits the generalization of our results to only PSAPs similar to the one used in our study. Several brands of PSAPs are on the market, ranging from a few dozen dollars to a few hundred dollars, with different features. The OTC market (OTC hearing aids and PSAPS) rapidly expands, providing many options when purchasing OTC amplifiers. It is urgent to conduct studies comparing other brands of devices, focusing on the features that best maximize the benefits, especially in adverse listening conditions. Noise suppression and directional microphones could be an essential feature, as difficulty listening to speech-in-noise is a frequently reported complaint in aging, even in older adults with normal hearing. We also did not compare the benefits of PSAPs with conventional hearing aids. PSAPs may provide significantly more benefits than the unaided condition but substantially less than traditional hearing aids, which is consistent with the results of a previous study. If this is the case, it would suggest that conventional hearing aids should be preferred over PSAPs for treating hearing loss. It is also possible that PSAPs and conventional hearing aids offer similar benefits, consistent with the meta-analysis mentioned above, indicating that PSAPs may be as effective as conventional hearing aids in treating hearing loss. Although this comparison is critical to understand the extent of the benefits of PSAPs, it is difficult to do given the many different types of hearing aids and PSAPs. Further studies should compare PSAPs and hearing aids with similar characteristics.

The lack of objective device output verification is another limitation of the study, as it raises questions about the reliability and accuracy of the data and may be related to the large individual differences in benefit from these devices. The study did not quantify or account for variations in the output of the devices in the ear canal or how well they were coupled to the ear. This factor could be an important predictor of interindividual variability in benefit and was not considered in the current study's design. Nevertheless, we have effectively overcome this issue by verifying the device's functionality with self-reported measures, ensuring a minimum level of amplification that participants found noticeable. This study is encouraging because it demonstrates that the participants were able to benefit from self-fitting the devices with very little guidance.

Another limitation of this study is the placement of speakers in corners instead of using a single forward-facing speaker, which may not fully reproduce the ecological nature of face-to-face conversations. In addition, the use of babble noise, while providing some background interference, may not fully reflect the complexities of natural listening environments. The use of a multi-speaker background noise, which more closely resembles the acoustic conditions of group conversations, might have been more appropriate for improving the ecological validity of the task.

Finally, our sample size is relatively small, which may have reduced the statistical power of our results. We relied on sample sizes from previous studies to determine our sample size because we could not perform a power analysis based on the information provided in those studies. To address this literature problem, we systematically reported the effect sizes of our analyses. This will allow future studies to calculate their sample sizes more accurately based on the effect sizes reported in the current study.

## Conclusion

In an environment where OTC devices, including PSAPs, are becoming more widespread and readily available, our results are important for three reasons. First, we showed that PSAPs can help people with different hearing profiles, particularly those with higher hearing thresholds, to improve their listening ability compared with an unaided situation. Second, we demonstrated that adaptation to the devices is rapid and that benefits are apparent only after a few hours of use. Longer use is likely to yield greater benefits. Finally, we observed that not all users initially experience the same level of benefit, which varies with age, hearing status, and cognition. This reveals the importance of considering demographic and health factors to fully understand the benefits of hearing amplification. Previous studies have not included demographic and health factors in the analyses. Therefore, this may explain why some studies have favoured PSAPs over conventional hearing aids, and others have favoured conventional hearing aids over PSAPs. However, it needs to be clarified whether these factors only contribute to rapid acclimation to the device or whether they also indicate which users will experience long-term benefits.

## Supporting information

**S1 File. CONSORT checklist.**
(DOCX)

**S2 File. Study protocol.**
(DOCX)

## Acknowledgments

We thank William Ji for his contribution to testing and data collection. We also thank all participants for their time and commitment.

## Author Contributions

**Conceptualization:** Maxime Perron, Claude Alain.

**Data curation:** Maxime Perron.

**Formal analysis:** Maxime Perron.

**Funding acquisition:** Claude Alain.

**Investigation:** Maxime Perron, Brian Lau.

**Methodology:** Maxime Perron, Claude Alain.

**Project administration:** Maxime Perron.

**Resources:** Claude Alain.

**Supervision:** Claude Alain.

**Visualization:** Maxime Perron.

**Writing – original draft:** Maxime Perron.

**Writing – review & editing:** Brian Lau, Claude Alain.

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
