## [Decision Letter · Decision Letter 0]

22 Mar 2023

PONE-D-23-03040Interindividual variability in the benefits of over-the-counter hearing devices on speech perception in noise: a randomized cross-over clinical trialPLOS ONE

Dear Dr. Perron,

Thank you for submitting your manuscript to PLOS ONE. After careful consideration, we feel that it has merit but does not fully meet PLOS ONE’s publication criteria as it currently stands. Therefore, we invite you to submit a revised version of the manuscript that addresses the points raised during the review process. As you can see from the reports included below, your manuscript has been assessed by three reviewers. They appreciated the timeliness of your research question, but they also raised several significant concerns that need your attention. For example, they noted overlapping concerns regarding the methodological reporting and the  statistical analysis of your data, which impact on the overall strength of your conclusions. Given the nature of the concerns raised, your revised manuscript will likely be sent to the original reviewers for re-assessment.     

We look forward to receiving your revised manuscript.

Kind regards,

Dario Ummarino, PhD

Senior Editor

PLOS ONE

Journal Requirements:

Reviewers' comments:

Reviewer's Responses to Questions

**Comments to the Author**

1. Is the manuscript technically sound, and do the data support the conclusions?

Reviewer #1: Partly

Reviewer #2: Partly

Reviewer #3: Partly

2. Has the statistical analysis been performed appropriately and rigorously? 

Reviewer #1: Yes

Reviewer #2: No

Reviewer #3: No

3. Have the authors made all data underlying the findings in their manuscript fully available?

Reviewer #1: Yes

Reviewer #2: Yes

Reviewer #3: No

4. Is the manuscript presented in an intelligible fashion and written in standard English?

Reviewer #1: Yes

Reviewer #2: Yes

Reviewer #3: Yes

5. Review Comments to the Author

Reviewer #1: Thank you for the opportunity to review this manuscript. The manuscript is well written and the research topic is timely. Please see my comment below for your consideration.

Major Concerns:

1. The introduction details quite nicely the impact hearing loss can have on individuals and the role hearing aids play in mitigating these issues. However, 13 of the 28 participants had normal hearing. It is unclear to this reviewer why participants with normal hearing were included given the rationale for the study related to mitigating hearing loss with low cost PSAPS. I suggest removing the normal hearing participants as I have difficulty understanding the interest in studying this group. If they are to be included, consider adding Group as a between subject variable in the statistical analyses to determine if the effects—although small—are different for normal versus impaired groups. By including the normal hearing participants in the current analysis, I suspect that the statistical power has been increased to the point where these small effects reach statistical significance.

2. Many studies investigating PSAPs and OTCs note that these devices are self-fitted by the individual, yet they typically involve assistance from the research team. This is a flaw of the current study as well. Page 11, lines 215-227 detail the fitting process of the device. Lines 220-221 suggest the participants were not guided or assisted and this fitting was typical of anyone that would use the device. The lines above suggest otherwise as these participants were fitted in a double-walled sound booth, earmolds were selected by the researchers, instructions and examples regarding the fitting were provided, additional assistance was provided if requested. It is unclear why the researchers did not provide the device and instructions as any person that purchased these devices on-line would receive, have them do an actual self-fit, then do the testing.

Other Concerns:

3. Page 8, participants, please include a power analysis to determine the appropriate sample size.

4. Figure 1 is not necessary.

5. Page 10, line 204-205, the rationale for using this device stated here is vague. What is this based on—‘good representation’? I suggest deleting this sentence. The sentence that follows makes a stronger rationale for selecting this device.

6. Page 11, lines 208-209, is this $300 cost per device or for 2 devices? Also, I searched this device on-line and it appears to be a BTE and not an in-ear device.

7. Page 11, line 210-211, consider changing this description to ‘dual directional microphone that is more sensitive to sounds from the front and less sensitive to sounds from the back.’

8. Page 11, line 224, was the NAL-NL2 fitting used based on the app? Is this the only option available or was this an option selected by the researchers for the participants?

9. Page 12, line 251, McArdle & Wilson (2006) determined the QuickSIN lists were not equivalent in difficulty for listeners with hearing loss and suggested lists 4, 5, 13 & 16 should not be used. List 4 was used in the current study.

10. Page 13, line 261, the example for identical is /tap/-/bat/ but these words are different.

11. Page 13, line 275, were the words presented through 2 speakers simultaneously? Why would you present the words from 2 speakers at +/-45 degrees as this isn’t a realistic situation.

12. Page 14, lines 290-293, why did you change from referring to the stimuli as CVC words to ‘syllables’?

13. Page 15, line 311, is Field a typo?

14. Page 20, line 398, why is there no result reported for the interaction of SNR x session?

15. Page 23, lines 478-479, see comment #1 in this review.

16. Page 24, lines 503-505, I am not sure the data support this statement, particularly the notion this could reduce the negative impact on cognitive performance. Also, a picky thought, but hearing aids do not restore hearing. Same on page 28, line 593.

17. Lastly, the manuscript is lengthy and has quite a bit of redundancies throughout. I get frustrated when a reviewer makes a comment that doesn't have a direct solution, so I realize this isn’t helpful. Just consider.

Reviewer #2: The aim of the study is to determine how Personal Sound Amplification Products (PSAPs) affect listening effort and speech perception in noise and measure interindividual variability and identify contributing demographic and health factors.

This is quite an interesting study, however, the manuscript requires improvement.

Comments

The following are to be provided/stated/described:

Line 154, the word abbreviation sd

Line 161, exclusion criteria

Line 168, the condition

Line 172, the second session.

Line 181, allocation concealment.

Line 179-182, information to be incorporated into Figure 1.

The study design not very clear and requires more description and with proper flow.

Line 208, $300 US to be written as USD300

Line 269, ‘((SNR; Pressuresignal/Pressurenoise) were used (SNR +3 dB, SNR 0 dB, SNR -3 dB).’ to be rewritten.

Line 311, typographical (Field[27].

Line 319, “very good,” “good,” “moderate,” “bad,” or “very bad” is to be presented as 'very good', 'good', 'moderate', 'bad' or 'very bad'.

Line 325, the title name is too short.

Line 341, the level of the accepted statistically significant to be stated.

Table 1, n to be stated. In the footnote, ‘with’ and ‘without’ to be denoted.

Range to be denoted as minimum to maximum.

Some results presented in the text are best presented in table form.

Line 367, the fulfillment of multiple stepwise regression analyses assumptions to be stated. Whether interaction and multicollinearity were explored is to be stated.

Line 469-470, the sentence to be rewritten.

Figure 1 to be revised based on the study design employed in the study (cross-over, assessment period, outcome measures etc to be stated.

List of references to conform to the journal format.

Reviewer #3: This manuscript describes an experiment where adults with sensorineural hearing loss were provided with personal sound amplification products. Speech recognition and self-reported listening effort were measured. Significant individual differences in speech recognition and listening effort were observed. Factors affecting individual differences in speech recognition were examined and age, hearing status and cognitive factors were found to influence benefit from PSAPs. The topic of this study is timely given the recent availability of over-the-counter (OTC) hearing aids in the United States. However, there were several significant limitations that would need to be addressed before the paper would be suitable for publication.

One major concern is the conceptual differences between the PSAP devices tested here and over-the-counter hearing aids. The title and various other parts of the paper describe PSAP and OTC devices interchangeably, but a more careful description of how the devices used in this study compare to OTC devices would be helpful. There is some overlap between these terms, but they are also distinctions that should be made so that readers understand the type of device that was evaluated and how that might compare to the more general category of OTC devices. Referencing the US FDA Final Rule on OTC definitions might help to provide some additional detail on how to classify this device using terminology that has recently become familiar to readers.

The description of the process of fitting the devices was confusing and replicating this study based on the details provided in the method would be challenging without further elaboration. Starting on line 215, the following questions should be addressed:

-The is reference to an earmold in the fitting. An earmold is traditionally a custom mold fitted to the specific patient’s ear by taking an impression. A clear description of the device coupling to the ear should be made in terms of whether this was a custom fitting or if the fitting was non-custom using different sized domes or ear tips.

-It says that the patients self-fitted the device, but in one sentence (Line 218) instructions were given and in the next sentence (Line 220), it says that fitting assistance was not provided to simulate self-fitting at home. This should be clarified as it might impact how these results relate to current OTC devices that do not include professional guidance or instructions.

-The fitting procedure describes a hearing test, assumably through the device (in situ), but that is not clear.

-The device based the fitting on the NAL-NL2 prescriptive formulae, but in the previous paragraph the device is described as having three pre-sets. Thus, it is unclear whether the fitting was customized based on the hearing test or if the device selects one of the presets closest to NAL-NL2 based on the hearing test. This is important because fitting to NAL-NL2 targets using only three presets for the range of hearing losses included in the sample would be difficult.

-Was the fitting verified in any way using probe microphone or electroacoustic measures to determine whether the devices provided consistent audibility or how closely they matched the NAL-NL2 targets?

The lack of verification of the output of the devices is a major limitation of the experimental design. The large individual differences in benefit from these devices could be related to variation in the output of the devices in the ear canal and variation in how well they were coupled to the ear. It is true that real-ear verification or coupler measures would not be available if these devices were self-fitted at home as PSAPs are, but the same argument could be made for all of the other outcomes in the experiment, and the lack of consideration for how these devices might have restored audibility for these participants is a significant limitation in that a key factor that could have determined the outcome was omitted from the design.

The use of a difference score in the statistical analyses, rather than creating a repeated-measures fixed effect for aided condition is problematic as the difference score ignores the variance of the unaided and aided conditions. The linear mixed effects models used here with random effects for each subject can account for the correlation between repeated measures within subject quite well, so it is unclear why the repeated measure was reduced to a difference score instead.

The stepwise regression approach based on AIC is concerning because the model comparisons and AIC criteria are not described. One can assume the mode with the best AIC was reported, but it is unclear how much better the final models were than comparator models for both speech recognition and listening effort. Much more detail about the statistical approach would be needed to interpret these models.

It is unclear whether the design has adequate statistical power to assess the number of predictors that were included in the model and whether the lack of power influenced the stepwise regression outcome.

The discussion includes overly general statements about potential benefits from PSAP and OTC devices that should be heavily qualified. This is an example where a clearer description of how these PSAP devices fit within the category of OTC devices would be helpful, as it could help readers to understand how the results with these devices might generalize to the broader landscape of OTC hearing aids.

A minor comment on the Introduction. The need for OTC devices in the introduction is heavily based on the cost of current prescription hearing aid being too high. It is worth acknowledging several studies that have shown that hearing aid adoption and hearing aid use are only marginally better in countries where prescription hearing aids are provided for free as part of government healthcare programs or at a significantly reduced cost. This should be acknowledged, and the revised discussion should include some consideration for how the benefit of OTC devices might compare to the benefit of prescription devices, if the latter were available at low or no cost.

6. PLOS authors have the option to publish the peer review history of their article (what does this mean?). If published, this will include your full peer review and any attached files.

Reviewer #1: No

Reviewer #2: No

Reviewer #3: No

---

## [Author Response · Author response to Decision Letter 0]

21 Apr 2023

Responses to reviewers are attached.

---

## [Decision Letter · Decision Letter 1]

13 Jun 2023

PONE-D-23-03040R1Interindividual variability in the benefits of over-the-counter hearing devices on speech perception in noise: a randomized cross-over clinical trialPLOS ONE

Dear Dr. Perron,

Thank you for submitting your revised manuscript to PLOS ONE and for your attention to the previous comments made by the reviewers. As you will see below, they still have some concerns. In particular the extent of assistance given to the participants, and the inclusion of those with normal hearing in the analysis. You do not seem to adequately address these, nor highlight them in the limitations section. After further  consideration, therefore, whilst we feel that your manuscript has merit, it does not fully meet PLOS ONE’s publication criteria as it currently stands. Therefore, we invite you to submit a revised version of the manuscript that gives further consideration to the points raised during the review process.

We look forward to receiving your revised manuscript.

Kind regards,

Antony Bayer

Academic Editor

PLOS ONE

Journal Requirements:

Reviewers' comments:

Reviewer's Responses to Questions

**Comments to the Author**

1. If the authors have adequately addressed your comments raised in a previous round of review and you feel that this manuscript is now acceptable for publication, you may indicate that here to bypass the “Comments to the Author” section, enter your conflict of interest statement in the “Confidential to Editor” section, and submit your "Accept" recommendation.

Reviewer #1: (No Response)

Reviewer #2: All comments have been addressed

Reviewer #3: All comments have been addressed

2. Is the manuscript technically sound, and do the data support the conclusions?

Reviewer #1: Partly

Reviewer #2: Partly

Reviewer #3: Yes

3. Has the statistical analysis been performed appropriately and rigorously? 

Reviewer #1: Yes

Reviewer #2: No

Reviewer #3: Yes

4. Have the authors made all data underlying the findings in their manuscript fully available?

Reviewer #1: Yes

Reviewer #2: Yes

Reviewer #3: Yes

5. Is the manuscript presented in an intelligible fashion and written in standard English?

Reviewer #1: Yes

Reviewer #2: Yes

Reviewer #3: Yes

6. Review Comments to the Author

Reviewer #1: Thank you for the opportunity to review this revised manuscript. The authors have not satisfactorily addressed the following concerns.

Major Concerns:

1. The authors added justification for including normal hearing participants in the study design. The justification is “PSAPs are designed for people that have difficulty hearing in certain situations, despite having normal hearing for their age.” They further noted “We focused on speech perception in noise because it is challenging for both older adults with and without hearing loss.” However, there is no information regarding the difficulties encountered by the normal hearing participants in the study. These were not normal hearing people with difficulties in noise based on the inclusion criteria. That being said, the normal hearing participants could have been evaluated separately from the hearing-impaired participants, as suggested, yet the authors did not do so. The effects observed were already very small. Including the normal hearing participants in the current analysis has increased the statistical power to the point where these small effects reach statistical significance. The authors suggest that using PTA as a continuous variable addressed my concern; however, the results showed session was related to PTA—further suggesting normal hearing participants aren’t candidates for these devices. Moreover, throughout the discussion, the authors seem to go back and forth between the findings being small and not clinically significant to PSAPs being a viable alternative to hearing aids.

2. As noted in the previous review, many studies investigating PSAPs and OTCs note that these devices are self-fitted by the individual, yet they typically involve assistance from the research team. This remains a flaw of the current study as well. In the revision, the authors note “No guidance (except if necessary) was necessary to participants during the test, as these devices are intended to be fitted at home without assistance of a healthcare professional.” How often was assistance necessary and to what extent? Again, it is unclear why the researchers did not provide the device and instructions as any person that purchased these devices on-line would receive, have them do an actual self-fit, then do the testing.

3. The test setup playing the target speech signal from 2 speakers at 45 degrees simultaneously is not realistic. The design does not, in any way, “mimic a complex and immersive listening environment, similar to those typically experienced by people with hearing loss, such as in group conversations or crowded restaurants, where sounds can emanate from all directions.” The use of a multi-talker babble, such as the one used in the QuickSin, would have been more appropriate.

4. In the discussion, line 543-546, again the authors are making claims about the value of PSAPs for normal hearing listeners that the data do not support as presented. The normal hearing data are not evaluated alone. The same, unfounded claim is made in the conclusions.

Reviewer #2: Minor comments:

Line 409, typo numerator.

Line 445 – 462, the results are to be presented in table form.

For the reduction of predictors in the analysis, the model fit is to be discussed.

Reviewer #3: The revision substantially addressed my comments from the original manuscript. I have two minor comments that should be easy to resolve.

-The fitting procedure has been substantially clarified in the revision. However, the method section now refers to the "NAL-NL2 procedure" when describing how gain for the devices is determined. There really is not an NAL-NL2 procedure, and if there were, if would suggest that the output of the devices was verified using traditional probe microphone measures. To avoid any confusion, I would state that the software set the gain prescription based on the results of the in-situ audiometric evaluation using the NAL-NL2 prescription.

-The term "over-the-counter devices" in the title should be changed to " a personal sound amplification product" since a single PSAP device was used, unless this device is classified as an OTC under the FDA definition and regulations.

7. PLOS authors have the option to publish the peer review history of their article (what does this mean?). If published, this will include your full peer review and any attached files.

Reviewer #1: No

Reviewer #2: No

Reviewer #3: No

---

## [Author Response · Author response to Decision Letter 1]

20 Jun 2023

The response to the reviewers is attached.

---

## [Editor Report · Decision Letter 2]

28 Jun 2023

Interindividual variability in the benefits of personal sound amplification products on speech perception in noise: a randomized cross-over clinical trial

PONE-D-23-03040R2

Dear Dr. Perron,

Thank you for your revised manuscript and for your further attention to the reviewers' comments. We’re pleased to inform you that your manuscript has been judged scientifically suitable for publication and will be formally accepted for publication once it meets all outstanding technical requirements.

Kind regards,

Antony Bayer

Academic Editor

PLOS ONE
---

## [Editor Report · Acceptance letter]

11 Jul 2023

PONE-D-23-03040R2 

Interindividual variability in the benefits of personal sound amplification products on speech perception in noise: a randomized cross-over clinical trial 

Dear Dr. Perron:

I'm pleased to inform you that your manuscript has been deemed suitable for publication in PLOS ONE. Congratulations! Your manuscript is now with our production department. 

Kind regards, 

on behalf of

Professor Antony Bayer 

Academic Editor

PLOS ONE